# Genome-Wide Identification of the 1-Aminocyclopropane-1-carboxylic Acid Synthase (*ACS*) Genes and Their Possible Role in Sand Pear (*Pyrus pyrifolia*) Fruit Ripening

**Jing-Guo Zhang** [1,2], **Wei Du** [2], **Jing Fan** [2], **Xiao-Ping Yang** [2], **Qi-Liang Chen** [2], **Ying Liu** [1], **Hong-Ju Hu** [2,*] and **Zheng-Rong Luo** [1,*]

[1] Key Laboratory of Horticultural Plant Biology, Huazhong Agricultural University, Wuhan 430070, China; zhjg@webmail.hzau.edu.cn (J.-G.Z.); liuying1991@webmail.hzau.edu.cn (Y.L.)

[2] Research Institute of Fruit and Tea, Hubei Academy of Agricultural Science, Wuhan 430064, China; duwei528529@126.com (W.D.); fanjing2013pear@163.com (J.F.); yangxiaoping1981@163.com (X.-P.Y.); cql99@sina.com (Q.-L.C.)

[*] Correspondence: hongjuhu@sina.com (H.-J.H.); luozhr@mail.hzau.edu.cn (Z.-R.L.)

**Abstract:** Ethylene production is negatively associated with storage life in sand pear (*Pyrus pyrifolia* Nakai), particularly at the time of fruit harvest. 1-Aminocyclopropane-1-carboxylic acid synthase (ACS) is the rate-limiting enzyme in ethylene biosynthesis and is considered to be important for fruit storage life. However, the candidate *ACS* genes and their roles in sand pear remain unclear. The present study identified 13 *ACS* genes from the sand pear genome. Phylogenetic analysis categorized these ACS genes into four subgroups (type I, type II, type III and putative AAT), and indicated a close relationship between sand pear and Chinese white pear (*P. bretschneideri*). According to the RNA-seq data and qRT-PCR analysis, *PpyACS1*, *PpyACS2*, *PpyACS3*, *PpyACS8*, *PpyACS9*, *PpyACS12* and *PpyACS13* were differently expressed in climacteric and non-climacteric-type pear fruits, 'Ninomiyahakuri' and 'Eli No.2', respectively, during fruit ripening. In addition, the expressions of *PpyACS2*, *PpyACS8*, *PpyACS12* and *PpyACS13* were found to be associated with system 1 of ethylene production, while *PpyACS1*, *PpyACS3*, and *PpyACS9* were found to be associated with system 2, indicating that these *ACS* genes have different roles in ethylene biosynthesis during fruit development. Overall, our study provides fundamental knowledge on the characteristics of the *ACS* gene family in sand pear, in addition to their possible roles in fruit ripening.

**Keywords:** sand pear; fruit ripening; ACS gene family; ethylene biosynthesis; climacteric; non-climacteric

## 1. Introduction

Ethylene is a phytohormone that participates in many events of plant growth and development, like organ abscission, stress responses, fruit ripening and senescence [1,2]. Generally, ethylene biosynthesis is divided into two systems, named system 1 and system 2. System 1 allows basal ethylene production in vegetative tissues and unripe fruit. System 2 involves the burst of ethylene production during the ripening of climacteric fruit. Moreover, system 1 is regulated in an auto-inhibitory manner, while system 2 is regulated in an autocatalytic manner. Fruit ripening behavior is generally characterized as being of a climacteric or a non-climacteric type, depending on whether or not the process is associated with a peak of ethylene production and a rise in respiration rate [3–5].

Ethylene is synthesized from *S*-adenosyl-L-methionine (SAM) via 1-aminocyclopropane-1-carboxylic acid (ACC). The two major processes in ethylene biosynthesis are the metabolic conversion of SAM to ACC by ACC synthase (EC 4.4.1.14) and secondly the conversion of ACC to ethylene by ACC oxidase (EC 1.14.17.4) [6]. In plant species, ACS is the rate-limiting enzyme in ethylene synthesis and is encoded by a widely diverse multi-gene family [7]. Ethylene biosynthesis is a complicated process involving numerous ACS genes.

The variable expression of ACS genes may be implicated in the synthesis of ethylene at a specific developmental stage. For example, there are 14 ACS genes in tomato (*Solanum lycopersicum*), among them *SlACS1A, SlACS2, SlACS4, SlACS11* and *SlACS12* showed different expression profiles during fruit ripening [8]. Three ACS genes (*MdACS1, MdACS3a*, and *MdACS6*) in apple have been extensively examined, and each of them was found to express distinctly [9–11]. *MdACS3a* was found to be expressed abundantly one month before fruit ripening, and its alleles are crucial in controlling fruit shelf-life, whereas *MdACS1* was shown to be responsible for the burst of ethylene production in system 2. *MdACS6* was expressed in the early stages of fruit development, before *MdACS3a* and *MdACS1* expression began [11].Among six *Pp-ACSs* isolated from the peach genome, *Pp-ACS1* and *Pp-ACS4* showed ripening-related increased expression during fruit development and ripening [12].

The sand pear (*Pyrus pyrifolia* Nakai) is widely distributed in southern China, where the ancestral pear species originated, and where a large collection of germplasm, including wild genotypes, landraces, and improved cultivars, exists [13,14]. Sand pear exhibits both climacteric and non-climacteric behavior depending on the cultivar or genotype, making it a suitable material for studying the fundamental basis of the various ripening behaviors [4]. In this study, the genome-wide analyses of ACS genes were conducted across five Rosaceae and two non-Rosaceae species. A total of 13 *PpyACSs* were identified in sand pear fruit and RNA-seq analysis showed the great variations between their expression levels. To further understand the role of *ACS* in fruit ripening, the predominantly expressed six ACS genes were selected and their expression profiles were compared in the fruits 'Ninomiyahakuri' and 'Eli No. 2' at different ripening stages. The findings of this study could help researchers better understand the molecular mechanism of fruit ripening in sand pear.

## 2. Materials and Methods

### 2.1. Plant Materials and Treatments

Two sand pear cultivars ('Ninomiyahakuri' and 'Eli No. 2') grown in Wuhan City, Hubei Province, P.R. China, were used in this study (Figure S1). Interestingly, both cultivars have different ripening behavior. Comparatively, fruits of 'Ninomiyahakuri' yield more ethylene during fruit ripening, whereas the fruits of 'Eli No. 2' yield less ethylene. The fruits of 'Ninomiyahakuri' and 'Eli No. 2' were collected 30 days before harvest (30 DBH; commercial maturity = 20 July 2017) and were kept at room temperature (24 °C) for 20 days. For lab analysis, fruits were sampled after an interval of 5 days after harvest (1, 5, 10, 15, 20 DAH). Five fruits were sampled at each sampling point for determination of ethylene production. Then the fruits were sliced, frozen in liquid nitrogen and stored at −80 °C for RNA extraction.

For spatial gene expression analysis, young fruit, mature fruit, flowers, leaf and stem were collected from 'Eli No. 2' pear trees. All samples were processed and stored as discussed above, for further analysis.

### 2.2. Measurements of Ethylene Production

The ethylene production was estimated with a gas chromatograph (Agilent 7890A, Santa Clara, CA, USA) fitted with a flame ionization detector. Briefly, fruits were placed in an airtight container (2.0 dm$^3$) equipped with septa and maintained at 24 °C for 1 h. Afterwards, 1 cm$^3$ of gas was sampled through the headspace of the container by a syringe.

### 2.3. RNA Isolation and RNA-Seq

Total RNA was extracted with TRIzol reagent (Invitrogen, Carlsbad, CA, USA) following the manufacturer's instructions. RNA concentration was measured by NanoDrop 2000 (Thermo, Waltham, MA, USA) and RNA integrity was evaluated by the RNA Nano 6000 Assay Kit using the Agilent Bioanalyzer 2100 (Agilent Technologies, Santa Clara, CA, USA).

Transcriptome libraries were constructed from samples of 'Ninomiyahakuri' and 'Eli No. 2' collected at three ripening stages: 1 (pre-ripening, 30 days before harvest), 2 (at harvest) and 3 (fruit senescence, 10 days after harvest). 1 µg RNA per sample was used for RNA-seq, which was carried out by Beijing Biomarker Technology Co., Ltd. (Beijing, China). In short, RNA-seq libraries were constructed using NEBNext®Ultra™ RNA Library Prep Kit for Illumina® (NEB, Ipswich, MA, USA) according to the manufacturer's protocol [15]. The cDNA library samples were sequenced by the Illumina Hi-Seq 2000 Sequencer. The transcriptome was aligned to the sand pear reference genome using Top Hat and the read count for each gene was obtained by the Cufflinks software. Quantification of gene expression levels were measured in fragments per kilobase of transcript per million fragments mapped (FPKM), and the DEGSeq R package (1.10.1) was used to analyze the differential expression of the pairwise comparisons [16].

### 2.4. Genome-Wide Identification of ACS Genes

In order to recognize the ACS gene family in *P. pyrifolia* and other four Rosaceae species (*P. bretschneideri*, *P. communis*, *Malus domestica* and *Prunus persica*), we searched the Genome Database for Rosaceae by BLASTP and BLASTN, using AtACSs sequences as queries [17,18]. The domain composition of ACSs were predicted by SMART (http://smart.embl-heidelberg.de/, accessed on 10 March 2021) and Pfam (http://pfam.san ger.ac.uk/, accessed on 10 March 2021). All of the identified ACS genes were analyzed using the NCBI's conserved domain database (https://www.ncbi.nlm.nih.gov/Structure/cdd/wrpsb.cgi, accessed on 12 March 2021) to determine the domains they possess. Multiple sequence alignment of ACS proteins was performed with Clustal X, the redundant or incomplete sequences were then removed. Additionally, ACS genes were named according to the guideline developed for Rosaceae family members [19].

The amino acid length and chromosome information were obtained from the genomic file. Protein molecular weights (Da) and isoelectric points (pI) were calculated by the online EXPASY server (http://web.expasy.org/protparam/, accessed on 12 March 2021). Gene motif analysis was performed with MEME v5.0.4 (http://meme-suite.org/tools/meme, accessed on 13 March 2021). Gene structure, motifs and chromosome location were presented using TBtools [20]. The Plant-mPLoc (http://www.csbio.sjtu.edu.cn/cgibin/PlantmPLoc.cgi, accessed on 15 March 2021) was performed to predict the subcellular localization of the ACS proteins. *Cis*-acting regulatory elements in the 2.0 kb of 5′-UTR of *PpyACS* were identified by the Plant-CARE database [21].

### 2.5. Phylogenetic Analysis

To identify orthologous genes, OrthoVenn2 was used to search against seven species genomes (Supplementary Table S1) using PpyACS as queries [22,23]. ACS amino acid sequences were aligned by using the MUSCLE program, and phylogenetic trees were built with the neighbor-joining method and 1000 bootstrap iterations in MEGAv7.0 software [24].

### 2.6. Quantitative Real-Time PCR (qRT-PCR) Analysis

The primers were designed by the Primer 6.0 program and listed in Table S2. Synthesis of the first-strand cDNA was conducted with PrimeScript™ RT reagent Kit with gDNA Eraser (TaKaRa, Shiga, Japan). cDNA was diluted 2-fold, and used as a template for qPCR. The PCR mix (50 mm$^3$) contained: 1.0 mm$^3$ Ex Taq and 25.0 mm$^3$ 2X Ex Taq Buffer (both Takara), 1.0 mm$^3$ dNTP Mix, 1.5 mm$^3$ of each primer (10X), 5.0 mm$^3$ cDNA, and 15.0 mm$^3$ PCR-Grade Water. The melt curve was assessed from 65 to 95 °C, with 0.5 °C increments. Relative levels of gene expression were calculated with the $2^{-\Delta\Delta Ct}$ method [25]. qRT-PCR gene expression was analyzed by CFX Connect™ (BIO-RAD, Hercules, CA, USA) using the PP2A gene as an internal control [26]. All analyses were achieved by three independent biological replicates.

## 3. Results

### 3.1. ACS Gene Sequence Identification

In this study, 56 ACS genes from pear were identified and it included 13 genes from apple and 8 from peach, 10 genes from Chinese white pear, 12 genes from European pear as well as 13 genes from sand pear. The detailed information of gene IDs, genomic positions, coding region lengths and translated protein sequence about pear ACS genes were performed in Table S3. Lineage-specific whole-genome duplication (WGD) was found to have occurred in the ancestor of pear and apple [27], which may have resulted in almost double the number of ACS genes in these four species than in peach. The numbers of ACS genes identified in three species of *Pyrus* spp. have slight differences that may be due to their genome assembly quality.

Based on the presence or absence of C-terminal phosphorylation motifs, ACS proteins are classified into three categories (type I, type II, and type III) [28]. Type I ACSs have a C-terminal segment with phosphorylation sites for both mitogen-activated protein kinases (MAPKs) and calcium-dependent protein kinases (CDPKs) (CDPKs). Type II isozymes only have the CDPK target site, but type III isozymes lack both the MAPK and the CDPK sites. There is also another ACS-like homolog, AtACS10 and AtACS12, which are thought to be amino acid transferases lacking ACS activity and are known as putative AAT [29].

All identified ACS genes contained a "1-Aminocyclopropane-1-carboxylic acid synthase" domain but belong to the different domain families: PLN02450 superfamily or PLN02450 exist in all types of ACS except type III, which specifically possess PLN2607. Interestingly, two type I ACS homologous genes (*AtACS1* and *AtACS2*) in Arabidopsis have a PLN02376 domain, which is not present in other investigated plants (Table S3).

In sand pear, all PprACSs contained the seven conserved regions of ACS (Figure 1) [23] and were located in 8 chromosomes unevenly: chromosomes 15 and 2 resided three genes, chromosomes 1 had two genes, and chromosomes 4, 6, 7, 8 and 14 only showed one gene (Figure S2). Subcellular prediction results showed that all PpyACS proteins were localized in the chloroplast, except for type II ACS, which were cytoplasm-localized proteins. The length of PpyACS-encoded protein sequences ranged from 445 (PpyACS11) to 611 (PpyACS13) amino acids. The predicted molecular weights and pI ranged from 49.84 kDa (PpyACS11) to 67.55 kDa (PpyACS13) and from 5.63 (PpyACS10) to 8.76 (PpyACS12), respectively (Table 1).

**Table 1.** Genetic bioinformation of all identified *ACS* genes in *Pyrus pyrifolia*.

| Gene Name | Gene ID | Chromosome | Protein Length (aa) | No. of Exons | Molecular Weight (kDa) | Isoelectric Point (pI) | Type | Subcellular Prediction |
|---|---|---|---|---|---|---|---|---|
| *PpyACS1* | Ppy15g2510.1 | Chr15:20022979..20024784+ | 473 | 4 | 53.24 | 6.47 | Type II | Cytoplasm |
| *PpyACS5* | Ppy02g1684.1 | Chr02:14156236..14158531+ | 473 | 4 | 55.54 | 6.9 | Type II | Cytoplasm |
| *PpyACS6* | Ppy08g0607.1 | Chr08:4476046..4478458- | 488 | 5 | 59.57 | 6.69 | Type II | Cytoplasm |
| *PpyACS7* | Ppy15g0540.1 | Chr15:3580798..3589288- | 502 | 5 | 54.67 | 7.06 | Type II | Cytoplasm |
| *PpyACS3* | Ppy01g0738.1 | Chr01:12651750..12654529- | 495 | 4 | 54.86 | 7.96 | Type I | Chloroplast |
| *PpyACS8* | Ppy07g1563.1 | Chr07:23041756..23045681+ | 529 | 5 | 53.18 | 8.53 | Type I | Chloroplast |
| *PpyACS9* | Ppy06g0809.1 | Chr06:12079561..12081861- | 487 | 4 | 56.82 | 7.56 | Type I | Chloroplast |
| *PpyACS4* | Ppy14g0933.1 | Chr14:11862299..11864545- | 487 | 4 | 55.15 | 6.64 | Type I | Chloroplast |
| *PpyACS2* | Ppy15g1745.1 | Chr15:12377889..12379594- | 446 | 3 | 50.1 | 5.65 | Type III | Chloroplast |
| *PpyACS10* | Ppy02g0565.1 | Chr02:3901240..3902919- | 447 | 3 | 50.31 | 5.63 | Type III | Chloroplast |
| *PpyACS11* | Ppy02g0566.1 | Chr02:3918274..3920013- | 445 | 3 | 49.84 | 5.84 | Type III | Chloroplast |
| *PpyACS12* | Ppy04g0675.1 | Chr04:6740597..6743142+ | 538 | 4 | 58.89 | 8.76 | Putative AAT | Chloroplast |
| *PpyACS13* | Ppy01g1669.1 | Chr01:20075291..20079590- | 611 | 6 | 67.55 | 8.3 | Putative AAT | Chloroplast |

The exon-intron organizations and motifs of all PpyACS genes were examined in *P. pyrifolia*. As shown in Figure 2, 16 conserved motif sequences were detected (Table S4). All PpyACS contained motif 1~10. Motif 11 and 12 were only found in type I ACS, whereas motif 15 was unique to putative AAT genes. Motif 13 was found in type I and type III ACS. In four type II ACS genes, motif 14 was present in PpyACS1 and PpyACS5, whereas motif 16 only existed in PpyACS1 and PpyACS5. In general, the motifs are quite conserved within each type. In addition, the presence of a varying number and length of introns

among the PpyACSs contribute to significant variations in gene length. The number of introns per PpyACSs also varied from two to five: the type III PpyACSs were characterized by two introns and the other PpyACSs displayed three to four introns except for PpyACS13, which contained five introns.

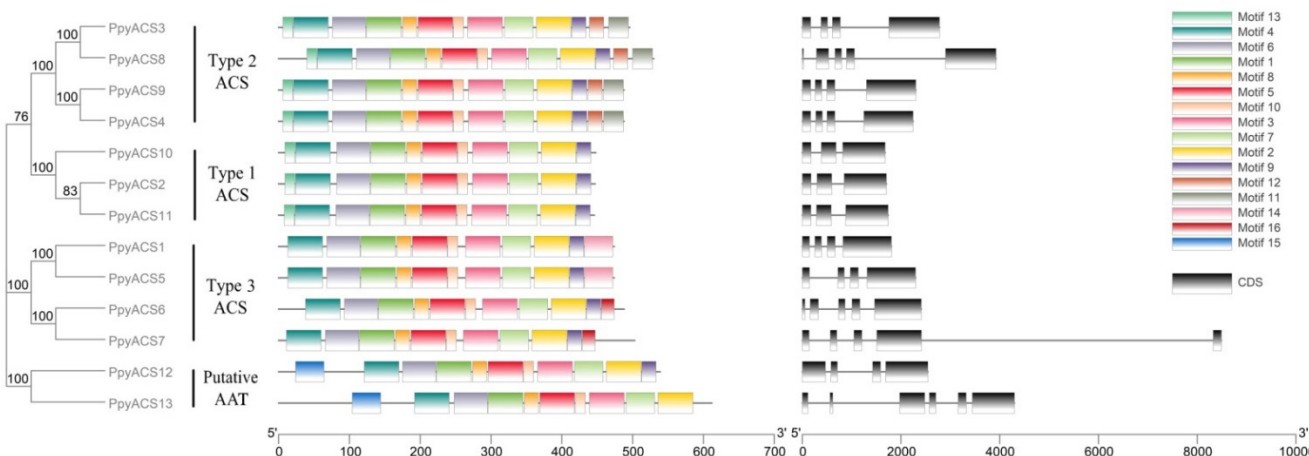

**Figure 1.** Amino acid sequence alignment of the PpyACSs. The seven conserved domains of the ACS isozymes are marked as red boxes. The seven highly conserved regions among all ACC synthases are underlined.

**Figure 2.** The phylogenetic relationship, conserved motifs and CDS structure of PpyACS proteins. A phylogenetic tree of the PpyACS protein family (**left side**) was constructed by MEGA7 using the Neighbor-Joining (NJ) method and 1000 bootstraps. The distribution of 16 conserved motifs across PpyACS protein (**middle side**) was predicted by the MEME (Multiple Em for Motif Elicitation) tools. The CDS structure of PpyACS genes were shown on the (**right side**). The sequences and lengths of motifs among PpyACS proteins were presented in Table S4.

To further understand the transcriptional regulation and potential functions of *Ppy*ACS genes, the upstream promoter sequences of *PpyACSs* (2000 bp upstream of the start codon) were isolated and predicted (Figure 3). A number of cis-elements implicated in hormone and stress responses were found in the upstream sequences of these *PpyACSs*. MeJA responsiveness elements (TGACG-and CGTCA-motif) were present in all *PpyACSs* except *PpyACS13*. ABRE, ABA-responsive element was identified in most of the *PpyACSs*, except *PpyACS4*, *PpyACS11* and *PpyACS12*. Three elements involved in gibberellin response, namely GARE-motif, TATC-box and P-box, were identified in all *PpyACSs*, except *PpyACS7*. Auxin-responsive elements (AuxRR-core and TGA-element) were found in most of the *PpyACSs*, except *PpyACS6*, *PpyACS7*, *PpyACS8* and *PpyACS11*. We also noticed that cis-elements associated with abiotic stress response were widely distributed within the promoters of *PpyACSs*. For example, an element essential for the anaerobic induction, named ARE, was identified in the *PpyACSs*, except *PpyACS6* and *PpyACS12*. A low temperature-responsive element (LTR) was identified in *PpyACS1*, *PpyACS3*, *PpyACS9* and *PpyACS10*, whereas TC-rich repeats involved in defense response were observed in *PpyACS1* and *PpyACS2*. Meanwhile, a MYB transcription factor binding site (MBS) involved in drought stress response was present in four members, including *PpyACS2*, *PpyACS7*, *PpyACS10* and *PpyACS11*. In addition, a wound-responsive element (WUN motif) was found in *PpyACS5*, *PpyACS9* and *PpyACS11*.

### 3.2. Phylogenetic Analysis of Putative ACS Genes

A total of 88 ACS proteins from seven species were phylogenetically categorized into four subgroups that perfectly fit their types (Figure 4). Type II included the highest number of ACS genes (27), followed by type I, which contained 25 ACS genes. This result provides further evidence that type I and type II contains a greater percentage of ACS genes to other types. Interestingly, each species has two members belonging to putative AAT that were presumed without ACS activity. In addition, from species phylogeny based on ACS orthologs genes (Figure S3, Table S5), a close relationship of ACS genes was observed between *Pyrus* and *Malus*, and the closest relationship was detected between *P. pyrifolia* and *P. bretschneideri* genes.

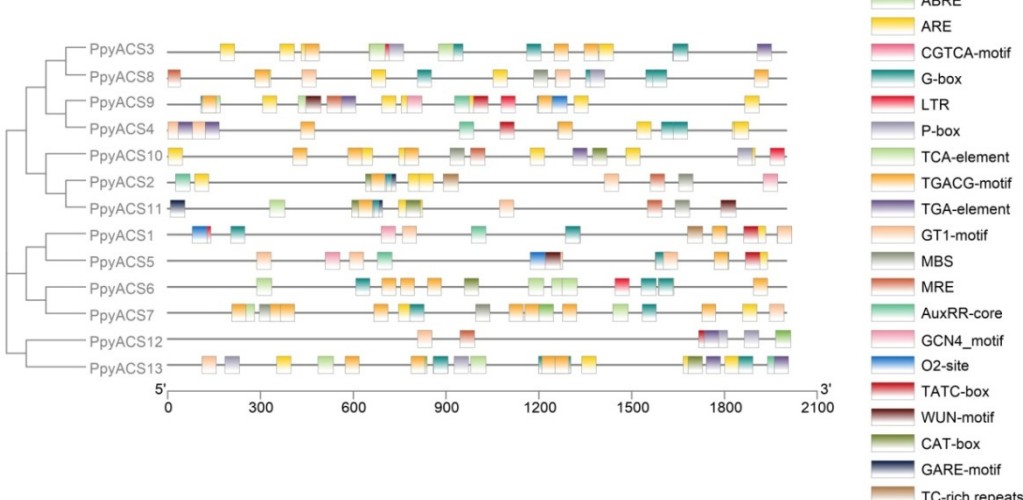

**Figure 3.** Cis-elements analysis of *PpyACS* genes. Note: *ABRE*, abscisic acid-responsive element; *ARE*, element essential for the anaerobic induction; *LTR*, low-temperature responsiveness; *MBS*, MYB binding site involved in drought-inducibility; *MRE*, MYB binding site involved in light responsiveness; *TCA-element*, salicylic acid-responsive element; *TGA-element,* auxin-responsive element; *TC-rich repeats*, defense and stress responsiveness; *TGACG-* and *CGTCA-motif*, MeJA responsiveness; *GARE-motif*, gibberellin-responsive element; *TATC-box*, gibberellin-responsive element; *WUN-motif*, wound-responsive element; *P-box*, gibberellin-responsive element; *AuxRR-core*, auxin-responsive element.

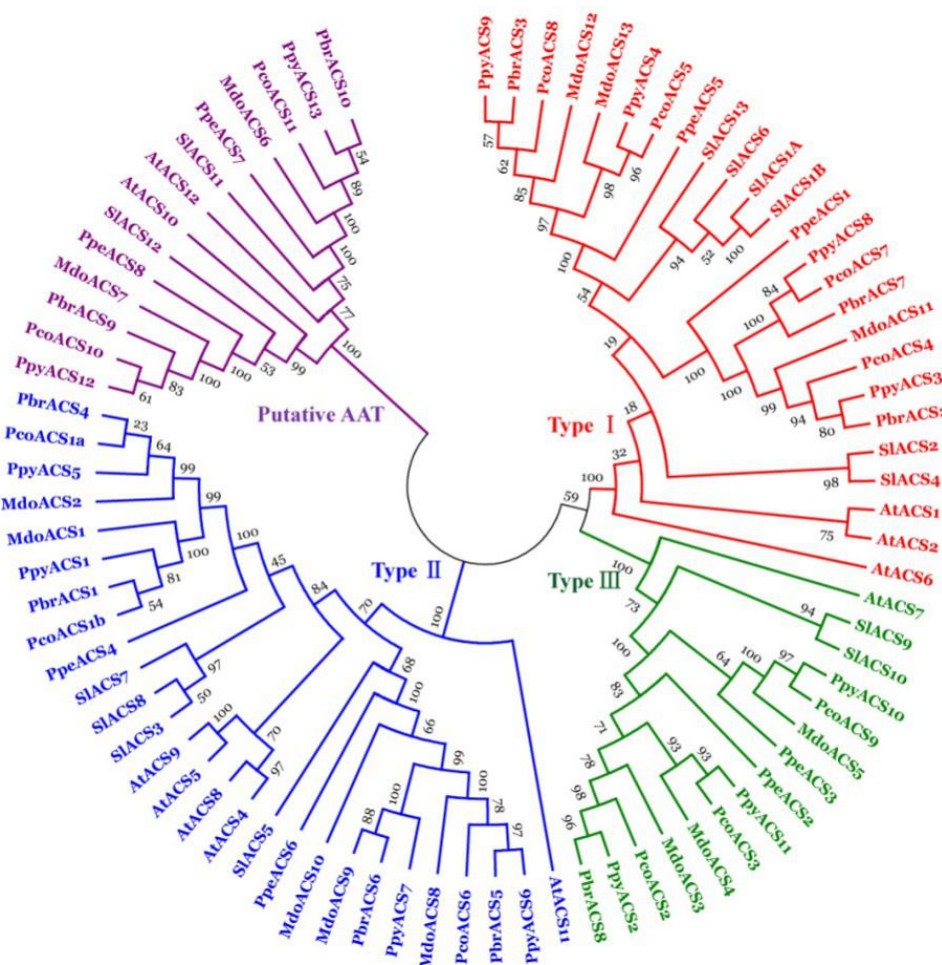

**Figure 4.** Phylogenetic analysis of ACS proteins from five Rosaceae species (*Pyrus pyrifolia*, *Pyrus bretschneideri*, *Pyrus communis*, *Malus domestica* and *Prunus persica*), and two non-Rosaceae species (*Arabidopsis thaliana* and *Solanum lycopersicum*). The phylogenetic tree was constructed using MEGA7.0 software by the neighbor-joining method. Different colors indicate different subfamilies of ACS.

### 3.3. Expression of PpyACSs in Different Organs

The qRT-PCR analysis showed that *Ppy*ACS genes were differentially expressed in different tissues. Among 13 *Ppy*ACS genes, *PpyACS1*, *PpyACS2*, *PpyACS3* and *PpyACS9* were expressed highly in mature fruits. The other nine *PpyACS* genes were expressed in the remaining studied tissues/organs, such as *PpyACS4* and *PpyACS5*, which were highly expressed in the leaf, while *PpyACS5*, *PpyACS7*, *PpyACS8*, *PpyACS10*, *PpyACS12* and *PpyACS13* in flowers (Figure 5).

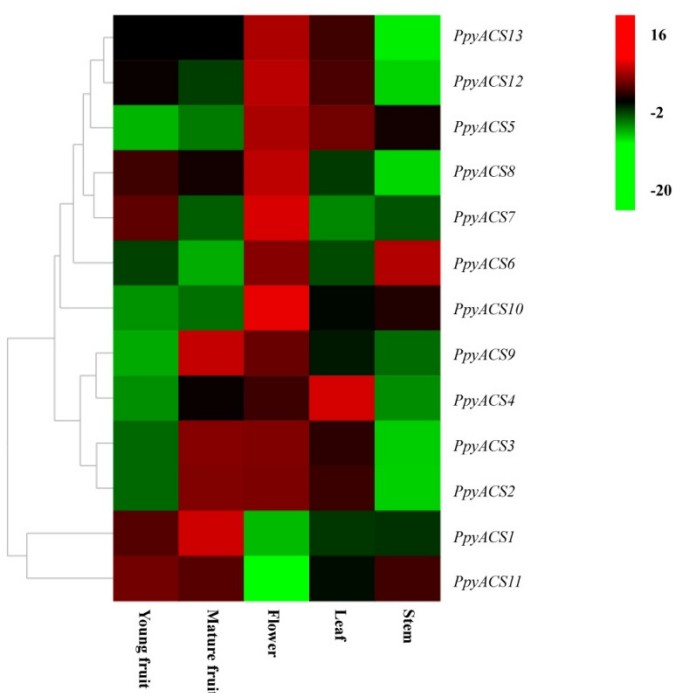

**Figure 5.** Heat map of the expression level of *PpyACSs* in different organs. The tissues used for the expression profiling are indicated at the bottom of each column. A cluster dendrogram is shown on the left. ACSs are divided into 3 major groups based on their expression. The color key at the top right corner represents the Z-score values transformed from log2-based expression values obtained by qRT-PCR.

### 3.4. RNA-Seq of Sand Pear Fruit and Identification of PpyACSs

In this study, six transcriptome libraries were constructed and sequenced for cv. 'Ninomiyahakuri' and 'Eli No. 2' in three ripening stages, including 1 (pre-ripening, 30 days before harvest), 2 (at harvest) and 3 (fruit senescence, 10 days after harvest). Over 89.2% of clean reads were uniquely mapped to the sand pear genome [30]. Expression of 13 *PpyACSs* in the pear genome was detected. The transcript abundance of *PpyACSs* varied greatly among members in different climacteric type sand pears, with that of one gene, *PpyACS1*, accounting for the most of total abundance in 'Ninomiyahakuri' (climacteric type) but none in 'Eli No. 2' (non-climacteric type). And six genes (*PpyACS2*, *PpyACS3*, *PpyACS8*, *PpyACS9*, *PpyACS12* and *PpyACS13*) were expressed during the fruit maturation in both cultivars but in different FPKM (Fragments per kilobase of transcript per million fragments mapped) (Figure 6). Based on these results, seven genes were selected for further analysis.

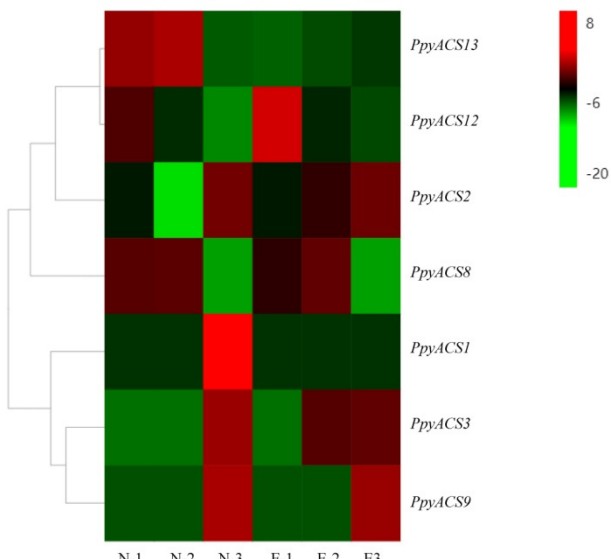

**Figure 6.** Heat map of the expression level of *PpyACSs* in fruit ripening. 'Ninomiyahakuri' and 'Eli No. 2' in three ripening stages, including 1 (pre-ripening, 30 days before harvest), 2 (at harvest) and 3 (fruit senescence, 10 days after harvest). A cluster dendrogram is shown on the left. ACSs are divided into two major groups based on their expression. The color key at the top right corner represents the Z-score values transformed from log2-based expression values obtained by RNA-seq (Table S6).

*3.5. Changes in Internal Ethylene Concentration in Flesh during Fruit Ripening*

To further elucidate the relationship between the expression pattern of the seven *PpyACS* genes and internal ethylene concentration, dynamic changes in internal ethylene concentration in the fruits of two cultivars were determined. As shown in Figure 7, there was a rapid increase in internal ethylene concentration and a large amount of ethylene was produced in 'Ninomiyahakuri' fruits that showed ripening behavior typical to climacteric fruits. In contrast, almost undetectable ethylene production was observed in 'Eli No. 2' fruits during postharvest.

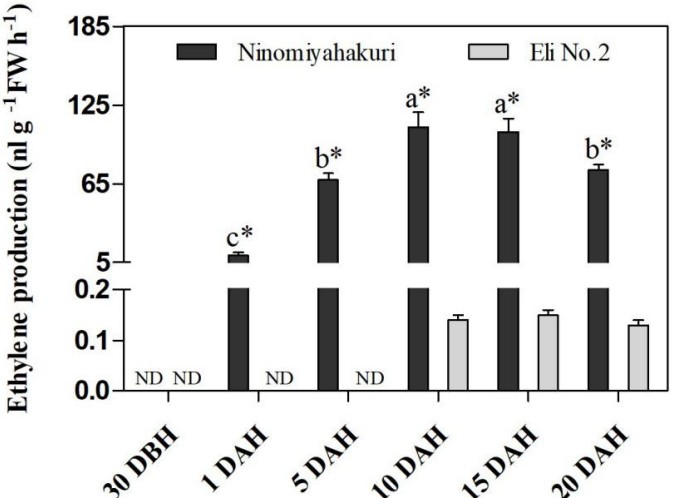

**Figure 7.** The ethylene production of fruits in ripening period of 'Ninomiyahakuri' and 'Eli No. 2' fruits. DBH: day before harvest. DAH: day after harvest. ND: None detected. '*' in each graph indicates a significant difference between 'Ninomiyahakuri' and 'Eli No. 2' cultivar (*t*-test, *n* = 3, *p* < 0.05). Different lower-case letters indicate significant differences among time points in the same cultivar (Duncan-test, *n* = 3, *p* < 0.05).

### 3.6. Expression of PpyACSs during Fruit Ripening

To validate expression profiles of the ACS genes isolated from transcriptome analysis, the seven *PpyACSs* were selected for qRT-PCR testing in fruits of cv. 'Ninomiyahakuri' and 'Eli No. 2' (Figure 8). *ACS1* was dramatically induced by ripening in 'Ninomiyahakuri' and showed the highest expression during postharvest, but was undetectable throughout ripening in 'Eli No. 2', which means *PpyACS1* is the dominant gene that regulates ripening-associated ethylene production in climacteric fruit. The expression of *PpyACS2* was also strongly increased during storage, not only in climacteric fruits but also in non-climacteric varieties, although the peak was approximately 2-fold lower than that in climacteric fruits. It is worth noting that three out of four type I genes have shown distinct expression patterns in fruit ripening, such as *PpyACS3* and *PpyACS9*, which were highly expressed after harvest in climacteric fruits, but the expression level of *PpyACS8* was higher at the pre-harvest stage than at the post-harvest stage in both climacteric and non-climacteric fruits. During storage, the expression level of *PpyACS8* decreased first and then increased. Interestingly, *PpyACS12* and *PpyACS13*, two ACS homologues of *AtACS10* and *AtACS12* which are presumed as amino acid transferases without ACS activity, were highly expressed at 30 days before commercial harvest and the expression level decreased slightly after harvest.

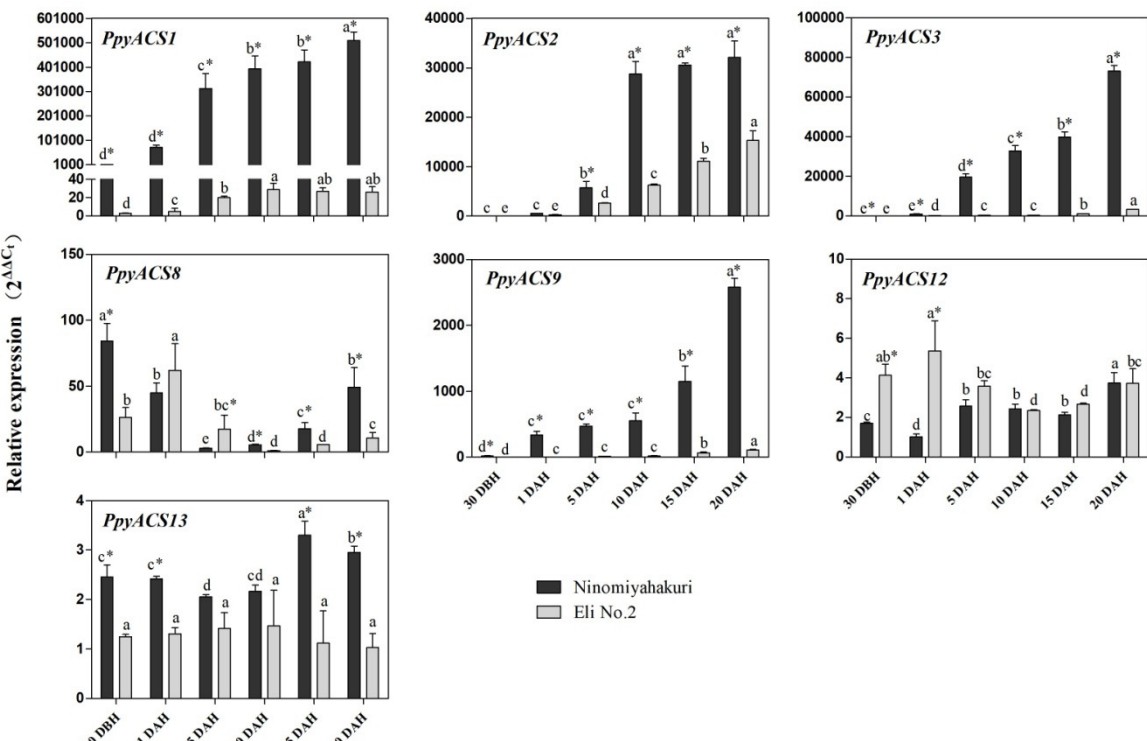

**Figure 8.** Comparison of *PpyACS1*, *PpyACS2*, *PpyACS3*, *PpyACS8*, *PpyACS9*, *PpyACS12* and *PpyACS13* expression levels between 'Ninomiyahakuri' and 'Eli No.2' fruits at different ripening stages. DBH: day before harvest. DAH: day after harvest. '*' Each graph indicates a significant difference between 'Ninomiyahakuri' and 'Eli No. 2' cultivar (*t*-test, *n* = 3, *p* < 0.05). Different lower-case letters indicate significant differences between time points in the same cultivar (Duncan-test, *n* = 3, *p* < 0.05).

### 4. Discussion

ACS is encoded by a multigene family with unique spatial and temporal expression patterns [8,12,23,31–33]. Although two ACSs have been found in sand pear [4], there are likely to be unidentified ACS genes in the pear genome, and their relationship with ripening behavior is unknown. Currently, the full-length genome of sand pear (*P. pyrifolia*) has been sequenced, providing valuable gene resources for further functional genomics study [30]. In this study, a total of 13 ACS genes were found in the genome of sand pear, and the

number is the same as that identified in *P. ussuriensis* [34]. A comprehensive overview of the ACS gene family was undertaken, including the gene structures, conserved motifs, chromosome locations, phylogeny, and *cis*-elements in the promoter sequence, as well as expression patterns.

It was noticed that the number of species in which the fruits of different varieties exhibit both climacteric and non-climacteric behavior and the classification of fruits into climacteric and non-climacteric categories are an over-simplification [35]. Consistent with previous reports [4,36], substantial differences in ethylene production during fruit ripening was observed between two sand pear cultivars, which indicated that sand pear shows climacteric as well as non-climacteric behavior, depending on the genotype. Previous research showed that ethylene production in climacteric sand pears is attributed to *PpyACS1* and *PpyACS2* expression during the ripening stage [4], but the mechanism of fruit ripening in non-climacteric fruit is unclear.

In this study, genome-wide expression analysis based on RNA-seq data and qRT-PCR was carried out with seven genes identified as the most strongly expressed genes during fruit ripening, *PpyACS1*, *PpyACS2*, *PpyACS3*, *PpyACS8*, *PpyACS9*, *PpyACS12* and *PpyACS13* (Figure 6, Table 2). *PpyACS1* was pre-dominantly expressed only in climacteric fruits after harvest and in accordance with the change in system 2 ripening-associated ethylene production, which indicated that *PpyACS1* may play key roles in the determination of ethylene production of climacteric fruit ripening. The same result was verified in previous studies [4,31,37,38]. Otherwise, as a type II gene, *PpyACS1* was probably under the regulation of ETO1 (ETHYLENE OVERPRODUCER 1) [28], which needs further research.

**Table 2.** Summary of *PpyACSs* gene expression during sand pear fruit development and ripening.

| Fruit Type | Main ACSs Involved in Ethylene Production | |
| --- | --- | --- |
| | Immature Stage | Mature Stage |
| Climacteric | PpyACS8, PpyACS12, PpyACS13 | PpyACS1, PpyACS2, PpyACS3, PpyACS9 |
| Non-climacteric | PpyACS8, PpyACS12, PpyACS13 | PpyACS2 |

Type III ACS proteins lack both MAPK and CDPK sites for post-translational regulation and may be of minor importance in fruit ripening [12]. However, a type IIIACS gene, *PPACS2* (*PpyACS2*), appears to be only expressed in the ripening of sand pear cultivars producing high or moderate levels of ethylene [4]. Surprisingly, our results showed that *PpyACS2* was expressed highly, even in non-climacteric fruit after harvest, which indicated that *PpyACS2* was responsible for system 1 ethylene biosynthesis in fruit ripening, similar to its ortholog gene, *MdACS3*, in apple.

Several investigations have revealed that type I functional divergence was the prevailing trend in the evolution of ACS subclades in plants [39]. In tomato fruit, it is well established that *SlACS6* and *SlACS1A* expression is associated with system 1 of ethylene production, while *SlACS4* and *SlACS2* are associated with system 2 [8,40]. Interestingly, all these four ACSs belong to type I (Figure 4) [41]. However, no type I ACS genes were expressed in apple fruit [33]. In the present study, three type I genes were newly identified—*PpyACS3*, *PpyACS8* and *PpyACS9*—and were involved in ethylene biosynthesis during fruit ripening. *PpyACS8* was highly expressed in the pre-ripening stage and is mostly responsible for the auto-inhibitory low levels of system 1, while *PpyACS3* and *PpyACS9* were highly expressed after fruit ripening and are associated with the burst of ethylene production in system 2. These results suggest that the type I ACS genes probably obtained unique and redundant functions during evolution.

It has been suggested that plant ACS genes evolved from ACS-like genes, which come from AATase genes [39]. The number and sequence of ACS-like genes are highly conserved in each species identified, which supported the ACS originate evolution of ACS-like genes. ACS-like genes, such as *AtACS10* and *AtACS12*, encode aminotransferases and without ACS activity, they were not generally considered to be authentic ACS genes. However, ACS-like genes in tomato, *SlACS11* and *SlACS12* display a significant up-regulation during

fruit ripening. Li et al. (2015) have reported that ethylene in young apple fruits is mainly produced by catalysis by *MdACS6*, an ACS-like gene that was mistaken for a type III gene [11,33]. In our data, the expression levels of *PpyACS12* and *PpyACS13* were low but expressed constitutively throughout the experimental period in both cultivars. Therefore, ACS-like gene expression patterns in tomato, apple and sand pear suggested that ACS-like genes are also responsible for the system 1 ethylene biosynthesis.

Analysis of the pear genome has revealed numerous duplicated genes (paralogs) forming large synteny blocks, covering major portions of chromosomes [27]. Interestingly, all PpyACS genes are distributed on chromosome pairs with partial synteny blocks rather than on chromosome pairs with more than 90% syntenic relations. For instance, *PpyACS1* (*Ppy15g2510.1*) and *PpyACS5* (*Ppy02g1684.1*) were paralogs, and have sequence similarities of more than 94%, located in the chromosome pairs with partial synteny blocks, Chr02 (upper) and Chr15 (middle-upper). The result indicated that *PpyACS* gene expansion resulted from the WGD event. Interestingly, both *PpyACS1* and *PpyACS2*, two key ACS genes that operate in system1 and system 2 ethylene biosynthesis in fruit ripening, respectively, were located in the middle of chromosome 15. These findings implied that this region is most important in the regulation of fruit ripening and therefore needs to be focused on in future investigations.

The localization of proteins in different organelles may be associated with their function [42,43]. Previous reports have indicated that ACS is localized in the cytoplasm [44–46]. Interestingly, subcellular localization predictions show that only type II ACS is located in the cytoplasm and other types of ACS in chloroplasts. Further subcellular localization testing is required to confirm this.

## 5. Conclusions

In conclusion, the genome sequencing data showed a total of 13 ACS genes in sand pear, nine of which were novel members. Seven of these ACS genes were found to be involved in sand pear fruit ripening, while four of them were found to be involved in system 1 ethylene biosynthesis, indicating that they play diverse roles in ethylene biosynthesis systems. This study also highlighted the need to include additional ACS genes in studies on ethylene production in sand pear fruits.

**Supplementary Materials:** The following are available online at https://www.mdpi.com/article/10.3390/horticulturae7100401/s1, Figure S1: Ripe fruit of 'Ninomiyahakuri' and 'Eli No. 2' pears, Figure S2: Distribution of ACS genes on the *Pyrus pyrifolia* chromosomes, Figure S3: Species tree of five Rosaceae species and two reference species. Table S1: ACS homologous genes identified in five Rosaceae species and two non-Rosaceae plant species, Table S2: Primers used for qRT-PCR analysis, Table S3: Detailed information of all ACS family genes identified in the seven species, Table S4: Sequences and lengths of motifs among PpyACS proteins, Table S5: ACS orthologs genes identified in the seven species, Table S6: Different expression of ACS genes in sand pear fruit ripening.

**Author Contributions:** Conceptualization, J.-G.Z. and Z.-R.L.; methodology, J.-G.Z., Y.L. and J.F.; formal analysis, J.-G.Z. and W.D.; investigation, J.-G.Z., H.-J.H., X.-P.Y. and Q.-L.C., writing—original draft preparation, J.-G.Z.; writing—review and editing, J.-G.Z., H.-J.H., W.D. and Z.-R.L.; supervision, J.F. and X.-P.Y.; project administration, J.-G.Z.; funding acquisition, H.-J.H. and Z.-R.L. All authors have read and agreed to the published version of the manuscript.

**Funding:** This research was funded by the China Agriculture Research System of MOF and MARA (CARS-28) and the Major Program of Hubei Agricultural Science and Technology Innovation Center (2020-620-000-002-05).

**Institutional Review Board Statement:** Not applicable.

**Informed Consent Statement:** Not applicable.

**Acknowledgments:** The authors thank Syed Bilal Hussain (University of Florida, USA) for peer review of the manuscript and for providing valuable comments and suggestions. We are also grateful for the patience and help of the editors and the reviewers.

**Conflicts of Interest:** The authors declare no conflict of interest.

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
