# Peer review of "Genome-Wide Identification of the 1-Aminocyclopropane-1-carboxylic Acid Synthase (ACS) Genes and Their Possible Role in Sand Pear (Pyrus pyrifolia) Fruit Ripening"

_horticulturae, doi:10.3390/horticulturae7100401_

Round 1

Reviewer 1 Report

All the articles consulted by the author should be listed under references including:

Liying Qi, Ling Chen, Chuansen Wang, Shaoling Zhang, Yingjie Yang, Jianlong Liu, Dingli Li, Jiankun Song and Ran Wang; Characterization of the Auxin Efflux Transporter PIN Proteins in Pear, Plants 2020, 9(3), 349; https://doi.org/10.3390/plants9030349

Yuan Gao, Jian Ma, Jia-Cheng Zheng, Jun Chen, Ming Chen,Yong-Bin Zhou, Jin-Dong Fu, Zhao-Shi Xu and You-Zhi Ma ; The Elongation Factor GmEF4 Is Involved in the Response to Drought and Salt Tolerance in Soybean; Int. J. Mol. Sci. 2019, 20(12), 3001; https://doi.org/10.3390/ijms20123001

Mudassar Ahmad, Jianzhao Li, Qinsong Yang, Wajeeha Jamil, Yuanwen Teng, and Songling Bai;  Phylogenetic, Molecular, and Functional Characterization of PpyCBF Proteins in Asian Pears (Pyrus pyrifolia); Int. J. Mol. Sci. 2019, 20(9), 2074; https://doi.org/10.3390/ijms20092074

Liqin, G., Jianguo, Z., Xiaoxia, L. et al. Polyploidy-related differential gene expression between diploid and synthesized allotriploid and allotetraploid hybrids of Populus. Mol Breeding 39, 69 (2019). https://doi.org/10.1007/s11032-019-0975-6

Wang, W., Wu, H. & Liu, JH. Genome-wide identification and expression profiling of copper-containing amine oxidase genes in sweet orange (Citrus sinensis). Tree Genetics & Genomes 13, 31 (2017). https://doi.org/10.1007/s11295-017-1102-7

There are numerous words and paragraph with additional space that need to be corrected.

e.g.

, Huazhong  Agricultural University   to , Huazhong Agricultural University

production  at the time  to   production at the time

etc. 

Remove all Paragraph Hyphenation. It makes it difficult to read

Fig 2. Difficult to read the notation on the phylogenetic tree. make the font bigger similar to Fig 3. 

Author Response

Thanks for your kind evaluation and comments on our paper. We have revised the manuscript according to your detailed suggestions. We sincerely hope this manuscript will be finally acceptable to be published on MDPI-MDPI-Horticulturae. 

Point 1: All the articles consulted by the author should be listed under references including:

Liying Qi, Ling Chen, Chuansen Wang, Shaoling Zhang, Yingjie Yang, Jianlong Liu, Dingli Li, Jiankun Song and Ran Wang; Characterization of the Auxin Efflux Transporter PIN Proteins in Pear, Plants 2020, 9(3), 349; https://doi.org/10.3390/plants9030349

Yuan Gao, Jian Ma, Jia-Cheng Zheng, Jun Chen, Ming Chen,Yong-Bin Zhou, Jin-Dong Fu, Zhao-Shi Xu and You-Zhi Ma ; The Elongation Factor GmEF4 Is Involved in the Response to Drought and Salt Tolerance in Soybean; Int. J. Mol. Sci. 2019, 20(12), 3001; https://doi.org/10.3390/ijms20123001

Mudassar Ahmad, Jianzhao Li, Qinsong Yang, Wajeeha Jamil, Yuanwen Teng, and Songling Bai;  Phylogenetic, Molecular, and Functional Characterization of PpyCBF Proteins in Asian Pears (Pyrus pyrifolia); Int. J. Mol. Sci. 2019, 20(9), 2074; https://doi.org/10.3390/ijms20092074

Liqin, G., Jianguo, Z., Xiaoxia, L. et al. Polyploidy-related differential gene expression between diploid and synthesized allotriploid and allotetraploid hybrids of Populus. Mol Breeding 39, 69 (2019). https://doi.org/10.1007/s11032-019-0975-6

Wang, W., Wu, H. & Liu, JH. Genome-wide identification and expression profiling of copper-containing amine oxidase genes in sweet orange (Citrus sinensis). Tree Genetics & Genomes 13, 31 (2017). https://doi.org/10.1007/s11295-017-1102-7

Response 1: All above articles has been included in the reference as your suggestions.

Point 2: There are numerous words and paragraph with additional space that need to be corrected.

e.g.

, Huazhong  Agricultural University   to , Huazhong Agricultural University

production  at the time  to   production at the time

etc. 

Remove all Paragraph Hyphenation. It makes it difficult to read

Response 2: The words and paragraph with additional space has been corrected, and all paragraph hyphenation was removed.

Point 3: Fig 2. Difficult to read the notation on the phylogenetic tree. make the font bigger similar to Fig 3. 
Response 3: We made the font of phylogenetic tree in Fig 2.bigger similar to Fig 3. 

Reviewer 2 Report

The manuscript by Zhang et al. aims at genome-wide identification and analysis of the ACS genes in Sand pear and determine their possible roles in fruit ripening. ACS, which have different roles in the biosynthesis of ethylene during fruit development were analyzed and used for the expression profiling at different fruit ripening stages.

This paper did a great job at providing information on the conserved motifs present, location on the gene, their gene structures, phylogeny, cis-elements and the overall expression patterns. The genome-wide expression profiling using RNA-seq analysis and qRT-PCR provided a list of seven most strongly expressed genes during fruit ripening and four involved in system 1 ethylene biosynthesis. Methodologically, this research sounds good. All the necessary considerations were taken while carrying out the experiment. The results were adequately discussed in the paper. Overall, this seems to be a perfectly composed manuscript ready for publishing.

Author Response

Thanks for your kind evaluation and positive comments on our paper. We carefully checked our manuscript again and revised to raise the manuscript level. We appreciate your contribution to this article.Your critical revision improved the standard of our article for publication and sincerely hope this manuscript will be finally acceptable to be published on MDPI-Horticulturae.

Reviewer 3 Report

Dear Authors,

Reviewer comments horticulturae-1399123

The manuscript entitled „Genome-wide identification of the 1-aminocyclopropane-1-carboxylic acid synthase (ACS) genes and their possible role in sand pear (Pyrus pyrifolia) fruit ripening“ represents a valuable complex molecular and physiological study aimed at a genome-wide identification, phylogenetic analysis, molecular characterization and expression analysis of crucial ethylene biosynthesis ACS genes between two sand pear cultivars, Ninomiyahakuri and Eli No. 2 as a climacteric one and a non-climacteric one. The manuscript provides original novel data in an appropriate and comprehensive manner thus providing complex information on ACS genes in two closely related pear cultivars differeing in their climacteric requirements.

I cant hus recommend the manuscript for publication; however, I have some comments which may improve the quality of the present manuscript:

Major comments:

Materials and methods:

SI units for volume: Use SI units for volume, i.e., use „dm3“ instead of „L“, „cm3“ instead of „mL“, and „mm3“ instead of „μL“, respectively.

Results and Discussion:

Phylogenetic tree analysis: In Figure2 and Figure 4 legends, the authors state that they used MEGA7 and neighbour-joining (NJ) method with 1000 bootstraps to construct the phylogenetic trees. However, in Figure 4 legend, the information on „1000 bootstraps“ is missing in the legend. I think that it should be added there. Moreover, in both phylogenetic trees, the bootstrap values have to be added at the individual nodes.

In Discussion, I recommend to add a comprehensive scheme as Figure 9 or a summarising table providing a comparison of the two sand pear cultivars,  Ninomiyahakuri and Eli No. 2 as a climacteric one and a non-climacteric one, respectively, with respect to ACS genes numbers, molecular characteristics and expression patterns to provide a comparison of ACS genes in closely related climacteric and non-climacteric pear species which will be useful for the readers.

Formal comments related to the text including typing errors, etc.:

Materials and methods, lines 78, 79: Remove the word „production“ (twice) in the sentence „…fruits of „Ninomiyahakuri“ yield more ethylene during fruit ripening, whereas the fruits of „Eli No. 2“ yield less ethylene.“

Materials and methods, line 128: Correct the typing error in the word „server“ („EXPASY server“) not „serve“.

Materials and methods, line 144, 2.6. heading: Correct the abberviation for quantitative real-time PCR as „qRT-PCR“ (not „qRT-PR“).

In Figure 4 legend, line 240, the authors have probably missed the information on the number of bootstraps used for the phylogenetic tree construction.

Results, line 310: Remove the word „of“ in the sentence „The expression of PpyACS2 was also strongly increased during storage not only in climacteric fruits but also in non-climacteric varieties….“

Results, line 317: Use a plural form in the words „two ACS homologues“.

Discussion, line 337: Add a space between „PpyACS1“ and „and“.

Discussion, line 388: Add a comme between the words „fruit ripening“ and „respectively.“

Final recommendation: Accept after a minor revision.

Author Response

Thanks for kind evaluation and valuable advises on our paper. We have revised the text and pictures carefully according to your comments. For example, we added a comprehensive summarising table (Table 2) as your suggestions. We tried our best to check the manuscript and corrected all errors. Hope this paper will be more qualified. Thanks again for your critical revision and it improved our manuscript quality for publication in MDPI-Horticulturae.

Dear Authors,
Reviewer comments horticulturae-1399123
The manuscript entitled „Genome-wide identification of the 1-aminocyclopropane-1-carboxylic acid synthase (ACS) genes and their possible role in sand pear (Pyrus pyrifolia) fruit ripening“ represents a valuable complex molecular and physiological study aimed at a genome-wide identification, phylogenetic analysis, molecular characterization and expression analysis of crucial ethylene biosynthesis ACS genes between two sand pear cultivars, Ninomiyahakuri and Eli No. 2 as a climacteric one and a non-climacteric one. The manuscript provides original novel data in an appropriate and comprehensive manner thus providing complex information on ACS genes in two closely related pear cultivars differeing in their climacteric requirements.
I cant hus recommend the manuscript for publication; however, I have some comments which may improve the quality of the present manuscript:
Major comments:
Point 1: Materials and methods:
SI units for volume: Use SI units for volume, i.e., use „dm3“ instead of „L“, „cm3“ instead of „mL“, and „mm3“ instead of „μL“, respectively.

Response 1: We revised SI units for volume as your suggestions.

Point 2: Results and Discussion:
Phylogenetic tree analysis: In Figure2 and Figure 4 legends, the authors state that they used MEGA7 and neighbour-joining (NJ) method with 1000 bootstraps to construct the phylogenetic trees. However, in Figure 4 legend, the information on „1000 bootstraps“ is missing in the legend. I think that it should be added there. Moreover, in both phylogenetic trees, the bootstrap values have to be added at the individual nodes.
Response 2: We added bootstraps in both phylogenetic trees as your suggestions.

Point 3: In Discussion, I recommend to add a comprehensive schemeas Figure 9 or a summarising table providing a comparison of the two sand pear cultivars,  Ninomiyahakuri and Eli No. 2 as a climacteric one and a non-climacteric one, respectively, with respect to ACS genes numbers, molecular characteristics and expression patterns to provide a comparison of ACS genes in closely related climacteric and non-climacteric pear species which will be useful for the readers.

Response 3: We added a comprehensive summarising table (Table 2) providing a comparison of ACS genes in closely related climacteric and non-climacteric pear species, as your suggestions.

Point 4: Formal comments related to the text including typing errors, etc.:
Materials and methods, lines 78, 79: Remove the word „production“ (twice) in the sentence „…fruits of „Ninomiyahakuri“ yield more ethylene during fruit ripening, whereas the fruits of „Eli No. 2“ yield less ethylene.“
Materials and methods, line 128: Correct the typing error in the word „server“ („EXPASY server“) not „serve“.
Materials and methods, line 144, 2.6. heading: Correct the abberviation for quantitative real-time PCR as „qRT-PCR“ (not „qRT-PR“).
In Figure 4 legend, line 240, the authors have probably missed the information on the number of bootstraps used for the phylogenetic tree construction.
Results, line 310: Remove the word „of“ in the sentence „The expression of PpyACS2 was also strongly increased during storage not only in climacteric fruits but also in non-climacteric varieties….“
Results, line 317: Use a plural form in the words „two ACS homologues“.
Discussion, line 337: Add a space between „PpyACS1“ and „and“.
Discussion, line 388: Add a comme between the words „fruit ripening“ and „respectively.“

Response 4: We seriously checked through our manuscript and correct all typing errors. 
Thanys for your suggestions.